# The Identification of Stakeholders' Living Contexts in Stakeholder Participation Data: A Semantic, Spatial and Temporal Analysis

Amal Marzouki [1,*], Sehl Mellouli [2] and Sylvie Daniel [3]

[1] Department of Information Systems, Université du Québec à Rimouski, Campus de Lévis 1595 Boulevard Alphonse-Desjardins, Lévis, QC G6V 0A6, Canada

[2] Department of Information Systems, Université Laval, 2325, rue de la Terrasse, Québec, QC G1V 0A6, Canada; sehl.mellouli@fsa.ulaval.ca

[3] Department of Geomatics, Université Laval, 1055, Avenue du Séminaire, Québec, QC G1V 0A6, Canada; sylvie.daniel@scg.ulaval.ca

[*] Correspondence: amal_marzouki@uqar.ca; Tel.: +1-418-9288-182

**Abstract:** Stakeholders' Participation (SP) aims to involve stakeholders in decision-making processes about significant choices affecting their organizations, cities, or communities. Stakeholders' participation is maintained through SP processes (SPPs) that may be traditional (e.g., physical assemblies) or online (e.g., online forums). Whether traditional or online, the purpose of SPPs is to collect and analyze data in a way that it would bring a benefit to a given decision-making process. In SPPs, stakeholders try to communicate (a part of) their living contexts, i.e., to present their objectives, daily problems, intentions, and issues they are facing within their environment. A major challenge of decision-makers is then to ensure that the living contexts of stakeholders are considered in SPPs for an effective implementation of project and policies. This paper focuses on the specific issue of the "stakeholders' living context identification" and attempts to account for how stakeholders implicitly identify their living contexts in their SP comments. Based on a qualitative analysis of SP data from four case studies in two countries, this paper identified a set of semantic, spatial, and temporal patterns allowing to capture the stakeholders' living contexts in SPPs data. Moreover, a conceptual model emphasizing the importance for decision-makers to capture and understand semantic, spatial, and temporal dimensions in SPPs is proposed.

**Keywords:** stakeholder participation; stakeholders' living contexts; semantic data analysis; spatial data analysis; temporal data analysis; place-based knowledge

## 1. Introduction

Stakeholder Participation Processes (SPPs) aim to reinforce the engagement of stakeholders in decision-making processes about significant choices affecting, for example, their organizations, cities or communities. They are designated as "two-way dialogues" bringing several benefits compared to "one-way processes" [1]. Unlike the approaches where only decision-makers and experts identify the problem and the potential solutions, the involvement of various stakeholders through SPPs may lead to better decisions [2]. Stakeholder Participation (SP) in decision-making processes brings information from different stakeholders with a diversity of views, values and needs. Over recent years, cities, governments as well as other public and private organizations adopted SPPs to increase the effectiveness of their decision-making processes [3]. With the emergence of Information and Communication Technologies (ICTs), SPPs took new forms by the use of, for example, dedicated solutions or social media platforms, which led to the concept of electronic participation (e-participation). Even though e-participation gained much significance as a buzzword, it maintains the same goals of participation in its traditional form that are increasing the

involvement of stakeholders and helping them achieve their communities' objectives [4,5]. Beyond the necessity to use various types of technologies to achieve participation goals, effective SPPs "are grounded in analyzing the context closely" [6] (p. 2). Indeed, one of the key issues identified in the literature is that SPP are, in some cases, disconnected from "stakeholder needs, preferences and priorities" and, therefore, lacking responsiveness to their "living contexts" [7,8]. In SPPs, stakeholders try to communicate issues that are [part] of their living contexts, i.e., to present their objectives, daily problems, intentions, and issues they are facing within their environment [9,10]. A major challenge of decision-makers is then to ensure that the living contexts of stakeholders are considered in SPPs for an adequate comprehension of the stakeholder inputs and consequently for effective decisions in project and policies implementation [6,10,11]. Hence, it becomes important to develop tools and techniques that help decision-makers capture and understand these living contexts. However, capturing the information about the living contexts is challenging as this information is implicitly expressed in SPPs data. This study takes the first steps towards understanding what characterize the living context of a stakeholder in SPP data by answering the following research question: how to identify stakeholder living contexts in SPPs and what patterns do they use to represent these contexts in SPPs inputs? In the context of this research, we will focus on textual inputs. Following a qualitative approach, we will investigate in this study three dimensions of the living context that are: semantic, spatial, and temporal dimensions. We will analyze comments (inputs) of SPPs according to these three dimensions to identify patterns related to each dimension. These comments are collected from four different case studies. Our findings show that, when extracted from data, these patterns help to capture the living contexts, enabling semantic, spatial, and temporal contextualization of SPP data. Moreover, the findings highlight that the three dimensions are not independent from each other, but they are interrelated. The rest of this paper is organized as follows. Section 2 provides the theoretical background for our research. In Section 3, we outline our research design and follow it with a description of the research methodology and implementation. Section 4 presents our research findings. In Section 5, we emphasize our research outcomes, which include the emergent conceptual model. In Section 6, we present the theoretical implications, we conclude our research, and we provide limitations and future research avenues.

## 2. Theoretical Background

### 2.1. Stakeholder Participation

Stakeholder participation is defined as "the practice of consulting and involving members of the public in the agenda-setting, decision-making, and policy-forming activities of organizations or institutions responsible for policy development" [11] (p. 512). It is one among several mechanisms that are used to involve stakeholders or their representatives in decision-making processes [9]. Stakeholder participation is viewed from the perspective of who the stakeholders are, how the stakeholders are represented, why the stakeholders are involved and what the stakeholders are involved in [6].

Generally, participation initiatives are of two types: spontaneous or solicited. Spontaneous participation consists of the spontaneous willingness of stakeholders to express their opinions or give suggestions for any organization (it could be a city, employer, government, etc.). Stakeholders can do it through different channels: physical or electronic forums, social media channels etc. For example, several cities around the word collect and analyze participation data, periodically, through their dedicated social media pages with the aim to enhance their efficiency and provide innovative plans to help address major urban strategic planning problems [12,13]. Solicited participation consists of a more formal way of participation where different phases of a SPP are planned and a definite duration is fixed. For example, a government can initiate a participation campaign for a specific project and in a specific period of time to involve stakeholders about significant decisions concerning their communities. Whether solicited or spontaneous, the purpose of stakeholder participation is to collect and analyze data (inputs) in a way that it would bring a benefit to

a given decision-making process, whatever its complexity. Hence, organizations should ensure that, the collected data is well understood to help provide decision-makers with the right information for an informed decision-making. Moreover, this information should be related to stakeholders' daily problems and priorities and depict their collective goals and intentions [13] and could be therefore considered as a reliable source to understand their living contexts.

*2.2. The Living Context and Contextualization in Stakeholder Participation*

The living context is defined in [14] as "the information about local issues, the topics related to everyday life" and "the information relevant to individual stakeholder" that "directly affect stakeholders' lives". It must be considered to better respond to stakeholders' requests in a participation process. The analysis of stakeholder participation in local governments showed that there is "a demand from the citizens' side to more effective communication about topics related to everyday life in their municipalities" [14] (p. 59). Indeed, the information about the living context (local issues, topics related to everyday life) and the information relevant for the stakeholder are considered as one of the most important communication needs in stakeholder participation [15]. When topics discussed are "distant from people's daily problems and priorities" [11] (p. 2), a SPP becomes limited and is below the initial expectations of organizations. This is consistent with the findings of [8,16] stipulating those projects and policies emphasizing the importance of capturing context-specific contingencies, as driven by stakeholders' voices, can be more effectively implemented when room for interpretation and discretion is given to stakeholders. Thus, stakeholders and decision-makers need to be aware and share a common understanding of their living-contexts to ensure effectiveness in decision-making. This context awareness has the potential to improve problem solving processes, leading to a more effective implementation of projects and policies. However, the context is continuously changing and evolving over time. What is a collective need or priority for stakeholders today may change, evolve or no longer be a need or a priority in the medium or long term [17]. Thus, stakeholders as well as decision-makers should be able to capture this change and to update their understanding to adequately meet the evolving needs of their communities.

The idea of contextualization aims to make explicit the living contexts that stakeholders express implicitly in SPPs. Three dimensions could characterize the living contexts: semantic, space and time. First, the spatial dimension is very important to consider when we retrieve the living contexts in SPPs as more than 80% of participation data has a geospatial reference [17,18]. The spatial dimension answers the question "where". In this sense, the spatial dimension provides an intuitive way to represent objects or events in a geographic space, allowing, among other things, the localization and the visualization of these objects or events. Objects with spatial dimension can be elements of our environment, such as natural geography objects (e.g., lands, vegetation, water, etc.) or human geography objects (e.g., roads, buildings, places, points of interest, etc.). Spatial dimension is often connected to the temporal dimension because spatial issues may change over time: "information on space-time changes can be an important asset for a successful SP" [17] (p. 1). Space and time are interconnected and depend on each other, and together they make the spatio-temporal dimension [19]. Hence, the second dimension to be considered in the living contexts is time.

Time is defined as the indefinite continued progress of existence and events in the past, present, and future [20]. The temporal dimension answers the question "when" [17]. Time may be represented and measured in seconds, minutes, hours, days, weeks and so on. Time can also be linear or a cyclic sequence [19,21]. Cyclic refers to iterations of events, such as the seasons [19,21]. The temporal dimension can be viewed as composed of two primitives: time points and time intervals [19]. A time point is an instant in time, and in contrast, a time interval is a temporal primitive with an extent. Beside spatial and temporal dimensions, there is also the semantic dimension.

The semantic dimension represents the meanings of the information that stakeholders give when they express their opinions during SPPs [22]. The semantic dimension answers the question "what are we discussing"? Combined with spatial and temporal dimensions, the semantic dimension generally refers to a theme or a topic to identify and describe concerns that can be spatially located (e.g., district, building, department, city etc.) and that may have a temporal characterization or evolve over time. The theme can for example represent human related concerns, such as social, political, demographical, or environmental concerns. Adding the semantic dimension to spatial and temporal dimensions brings a sense to what is discussed, making it more meaningful and improves the understanding of the stakeholders' living contexts [23,24].

*2.3. Semantic, Spatial and Temporal Data Analysis in Literature*

Semantic, spatial, and temporal dimensions have been studied and apprehended in different ways in several disciplines such as geomatics, linguistics, or computer science [25–29]. Hereafter, we briefly present relevant literature about semantic, spatial, and temporal dimensions analysis to consolidate the theoretical foundation of our research. Based on this literature, we present then our theoretical framework of semantic, spatial, and temporal SPPs data contextualization.

Semantic analysis in the literature: The semantic analysis of data consists of applying techniques and algorithms to depict topics from data. In computer science, most of semantic analysis methods apply algorithms based on machine learning and statistical techniques. For example, we can cite the use of unsupervised learning such as clustering algorithms to automatically detect topics within data [25,26]. Besides unsupervised learning techniques, semantic patterns can also be used to better interpret the data by, for example, extracting terms (nouns, adjectives, and verbs) from data and store them as nodes within a semantic network. Then, relations between terms can be represented. In this research, we aim to augment the foundation of existing techniques by identifying further semantic patterns in data to characterize the living contexts of stakeholders in SPPs comments.

Spatial analysis in the literature: We observe that there is no consensus on a given categorization of spatial entities in the literature [27]. In general, we distinguish between two main spatial concepts: objects and places [27,30]. Objects are "isolated material areas" that do not identify portions of space; they indicate the function of the object rather than its location [27,31]. For example, a wall is an object. On the other hand, places are entities fulfilling a localization function [30,32]. They are "purely spatial entities" that can be determined through their contours by means of spatial coordinates [33]. For example, a city is a place. The concept of place is based on the existence of a frame of reference that is a context or a point of view [1]. A frame of reference is defined as a "set of entities-places-endowed with spatial relationships that characterize their relative fixity during a given period and such that each determines an associated portion of space" [31]. Frames of reference help to identify spatial entities as places are characterized by their stability or fixity (in a given period) within an appropriate frame of reference or by portions of space in which target entities can be located [32,34]. In this study, we will limit the identification of spatial patterns in SPPs data to places or entities fulfilling a localization function.

Temporal analysis in the literature: Several lenses can be adopted to analyze temporality in texts [28]. According to [29], 'what happens psychologically in the case of time is the construction of a serial representation of events, processes and episodes ordered and/or anchored on the real time axis, on time axes in the future or on imaginary alternatives to the real time axis'. To perform this representation, means are needed to identify the related time axis and then to locate a moment, an interval, or an event. Two main temporal concepts can be considered to analyze temporality in texts: "the levels of analysis and representation" and "the temporal orders" [28,29].

According to the former, temporality in texts is understood at two main levels of analysis and representation: the first level refers to the task of anchoring temporal expressions (also called calendar expressions) in a calendar system (relating to "dates" or

"durations"); while the second refers to the task of calculating the temporal ordering of events in a text [28]. According to the latter (the temporal orders), four major orders exist for the apprehension of time in texts that are: modal, temporal, aspectual and enunciative orders [28]. Each of these orders asks the following questions:

- Modal order: is the content in the text presented as certain, possible, imaginary, etc.?
- Aspectual order: is the content presented as in progress or on the contrary as fully realized?
- Enunciative order: who is speaking? or who is presented as supporting such content?
- Temporal order: Is the content located in present, past, future time? What are its temporal coordinates?

In this research, we will apply the first level of analysis and representation of temporal expressions. To this end, we will identify and classify temporal patterns, mainly "temporal expressions" existing in SP data with the aim to highlight the temporal dimension of stakeholders' living contexts. Moreover, we will determine which temporal order enables the apprehension of the temporal dimension in textual SPP data.

### 2.4. Theoretical Framework—Semantic, Spatial and Temporal SPPs Data Contextualization

We present in Figure 1 the theoretical framework of SPPs data contextualization that will be used in this paper. Our theoretical framework suggests that SPPs data might be endowed with semantic, spatial, and temporal dimensions and if these dimensions are identified, it would offer a better understanding of the living contexts of stakeholders.

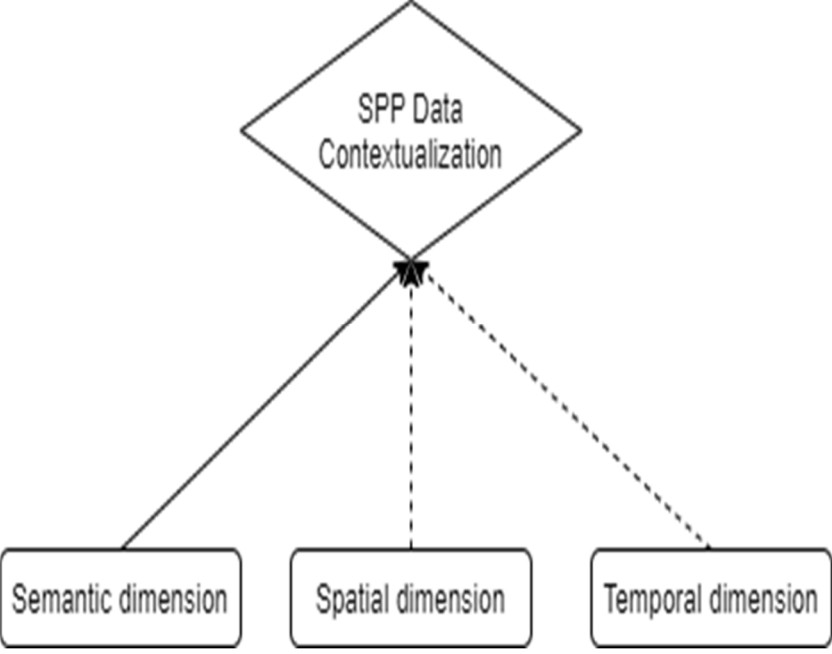

**Figure 1.** Theoretical framework—Semantic, spatial, and temporal SPP data contextualization.

In this framework, we consider that spatial and temporal dimensions are not mandatory (dotted lines). In fact, a stakeholder might not refer to spatial neither to temporal information in a SPP comment. However, the semantic dimension is mandatory since a SPP comment has necessarily a meaning to provide. In other words, a SPP comment can address a topic without as much addressing spatial and/or temporal information while a SP comment cannot address spatial and/or temporal information without addressing a topic.

## 3. Research Design, Methodology, and Implementation

The literature pointed to several studies related to SP focusing on different elements of participatory processes such as tools, engaging strategies, etc. [35,36]. However, there is still

a need for studies to inform how to take advantage from SPPs data [8,17]. Given this gap in the literature, the main research objective of this paper is to offer a better understanding of stakeholder living contexts in SPPs, through the identification of patterns that stakeholders implicitly use to represent these contexts in their SPP inputs. To meet this research objective, we will base our work on four different cases of SPPs in two different countries. This work will identify and categorize semantic, spatial, and temporal patterns that stakeholders use to represent their living contexts through their SPP comments. A pattern in natural language corpus analysis is defined as a regular and repeated using of words and their synonyms in "a way of deciding" that the usage of these words and their synonyms count as "a lexical meaning distinction" [37]. The patterns identified in this study are a regular and repeated way of using words or their synonyms that is formulated in a SPP comment.

### 3.1. Methodology

This research adopts a qualitative approach. Specifically, we adopt a multiple-cases design strategy. A case study "examines a phenomenon in its natural setting, employing multiple methods of data collection to gather information from one or a few entities (people, groups, or organizations). The boundaries of the phenomena are not clearly evident at the outset of the research and no experimental control or manipulation is used." [38] (p. 370). This research strategy is well aligned with our research objective. First, the use of a multi-case design is appropriate when a phenomenon is examined in a natural setting which is the case of our research. The data collected through the four cases is naturally occurring data, where the four processes of participation took place in their natural settings. Second, as new forms of SPPs are emerging and considered as contemporary phenomenon [38], a qualitative approach is appropriate as it allows better flexibility to explore the phenomenon under analysis allowing to adjust the whole data collection and analysis process [39].

### 3.2. Data Collection

Data collection depends on the research questions and the unit of analysis [38]. Multiple data collection methods are typically employed in research case studies. In this research, we explored four cases and we diversified the data collection methods. The collected data were a set of comments stated by stakeholders in four different SPPs with the intention to participate and to bring an opinion that would influence a decision. The first case study (case 1) consisted of a SPP for the strategic planning that was carried out in a public university between November 2017 and February 2018. The second case study (case 2) was a SPP that was held between 2015 and 2016 for the construction of a public square in a district in Canada. The third case study (case 3) concerned an SPP aiming to collect citizens' comments about a public collective transport company's service. The data was collected between March 2017 and April 2017. Finally, the fourth case (case 4) was an SPP aiming to collect citizens' comments about their city where the data was collected between January 2017 and December 2017. The three first cases took place in Canada while the fourth one was in Tunisia. The two first cases (cases 1 and 2) were solicited SPPs and the two last one (cases 3 and 4) were spontaneous SPPs.

The collected data came from four different sources: recorded and transcribed data, online form data; a participation platform data; Twitter data and Facebook data (see Appendix A). As our research objective was to study SPP data which consists of a set of comments provided by stakeholders, our unit of analysis was an expression in an individual SP comment.

### 3.3. Data Analysis

3.3.1. Data Analysis Process

Following the data collection, we proceeded with the data analysis. Two major iterations of data analysis were performed. In the first iteration, we analyzed the collected data of each case study. During the first iteration, we applied a structured multi-steps approach that enabled a constant comparison between the data and the emergent concepts [40].

Hereafter, we present the different steps of our approach. For the first case, we followed an a priori approach for data analysis [41] that we aligned with the three dimensions of our theoretical framework. First, for each dimension, we adopted an open coding approach which required a deep reading of the primary data to instill the data and to depict and understand the underlying concepts. Then, we conducted an axial coding to reveal relationships between first order concepts and second order concepts for each dimension. For each individual SP comment, we chose expressions as our unit of analysis to perform the coding; each expression in an individual participation comment that contained a pattern was added as an occurrence of that pattern. Finally, for the cases 2-3-4, we adopted an open coding approach with "Metacoding". Metacoding examines the relationship among a priori themes (already identified in open and axial coding of case 1) to discover potentially new themes and overarching meta-themes [41]. For each data unit, we looked at which patterns were identified, and which ones are emerging. Doing so, we were able to observe points of similarities and differences between the four cases. In the second iteration, we reviewed all the emerging patterns from the first iteration regarding the relevant literature on semantic, temporal and spatial analysis (see Appendix B). This second iteration aimed to find out whether there were patterns or models of patterns in the literature that are like those identified during the first iteration, in order to align the empirical results of our research with the existing models in the literature [42]. Only temporal patterns were adapted following the model of [28] that proposed a model of four calendar expressions. Indeed, five out of the six temporal patterns that we identified following the data analysis were similar to the calendar expressions of [28] (see Figure A1 in Appendix C). From these five patterns, two are subcategories of the same pattern of the model of [28], and the three others are similar to the other three patterns of the model of [28]. For semantic and spatial dimensions no similar patterns were found in literature.

### 3.3.2. Data Coding

Data coding was conducted combining different techniques as recommended by [41], mainly "repetitions", "word-synonyms co-occurrence" and "similarities and differences". The "repetitions" technique identifies expressions that "occur and reoccur" in SPPs comments [41]. "Word synonyms co-occurrence" identifies expressions that are "equivalent" [41] (synonyms) to other expressions and that can be classified in the same categories. "Similarities and differences" technique is a "constant comparison technique" that involves searching for similarities and differences by making systematic comparisons across units of data and cases [41].

In the first iteration, we interrogated the data using two 'seed' [42] questions: what are the patterns used by stakeholders to identify their living contexts? and do the identified patterns depend on the used tools (traditional or online)? The second question is mainly related to the data collected from the first case study where recorded and transcribed data come from a traditional (on-site) consultation and not from an online platform. Data coding was conducted following an iterative process of validation between the authors. The iterative process enabled refining our understanding of the identified patterns. We looked for expressions and meanings units that fit the pre-defined dimensions: semantic, spatial, and temporal. Starting with case 1, we used open coding where we applied the techniques of "repetitions" and "word synonyms co-occurrence" to identify patterns (see Table 1). Then, we applied an axial coding to build second order themes and evaluate patterns and their relationships. For cases 2–4, we not only applied the same open coding strategy used for case 1 but we also applied the third technique of "similarities and differences" to compare data units of the four cases. We carried out all the coding processes iteratively, by looking back to the case (s) to validate the outcomes of the process. The results of the coding process are presented in Table 1.

**Table 1.** Data structure: Semantic, spatial, and temporal patterns in SPP data.

| 1st Order Concept | 2nd Order Concept |
|---|---|
| Semantic Dimension | |
| There is a glaring lack of, there is no initiative for, the obstacles we see, it worries me that there is not, which poses a problem of | Issue |
| I think it would be interesting to, I make a proposal for, I would like to know if you would be ready, it will be interesting to ask, I think put more of, prove, and listen to them, Let us decrease the speed | Suggestion |
| I experienced this more than 20 years ago. We experience this everyday here, for having lived it for 2 years, this is my third year here, | Lived experience |
| In 20 programs, at 30 km/h, 300 employees, law project 21, $1 million, the 9-m rule, 36 buildings | Number/metric |
| United Nations, The Arctic council, The government of Quebec, World Health Organization, SPVM | Governing entity |
| Peter Simons, princess Lalla, Alexandre Tailleferre, Trump, Professor Sarah Woodruff | Reference |
| When will ? why ? where is? | Question |
| Bravo to the driver who kept her smile and was very patient, Thanks once again to the authorities for their responsiveness, | Compliment |
| I am attaching a small text which appeared recently in, http://dailynews.mcmaster.ca/smoke-free-campus-faq/ (accessed on 4 April 2022), I attach the document presented | Attached link/document (online only) |
| #worstsubwayever #polmtl #Transports #vivemtl #heuresdepointe #lignebleu #ariana #winou_etrottoir #Abaslacorruption #Urgent #corruption #douanetunisienne | Hashtag (online only) |
| @stminfo @stm_Orange@stminfo @CAA_Quebec @stm_nouvelles @stminfo @JourdelaTerreQc @SPVM @tvanouvelles | Tag_mention (online only) |
| 😊 😂 👍 | Emoticon |
| Spatial Dimension | |
| Senegal, Montreal, Maroc, Boston, France, Chad, City of Quebec, Cameroon, | Cities, Province and Countries |
| UQTR, USherbrook, Mc Master University, Western University | Similar organization with defined position |
| Institute EDS, Roger Van Den Hende botanical garden, PEPS, the department of Geography, Archeology and Anthropology | Internal entity with defined position |
| St-Louis-de-Gonzague college and Nazareth, The Museum of Civilization | External entity with defined position |
| West African countries, the organic community garden, North America, In the north | Spatial entity with approximated position |
| A 'mini-plant' for anaerobic digestion on campus, an outdoor ice rink which would be located between the De Koninck and Pouliot pavilions, places where we can make "power naps" | Spatial entity hypothetical position |

**Table 1.** *Cont.*

| 1st Order Concept | 2nd Order Concept |
|---|---|
| #ariana #sfax #menzah5 #communedelamarsa #kantaoui #tunis #zoo_tunis #parc_belvedere #municipalitédetunis #montreal | Spatial entity in hashtags (online only) |
| Ville de Quebec, Montreal, Gatineau, Gare de Vaudreuil | Spatial entity through location Stamp (online only) |
| Temporal dimension | |
| In September 2018, in January 2018, in 2020, by 2030, by 2050, | Future temporal expression |
| Since 1988, during fall 2015, during spring 2016, since January 2011, in 1999–2000, since 2004 | Past temporal expression |
| in the next 3–4 years, for almost 30 years, last year, 4 years ago | Temporal expression depending on the comment date |
| In 2017 later, In January 2016 two months later, In 2014 3 years before, | Temporal expression depending on another temporal expression in the comment |
| Since the second world war | Temporal expression recognized around the world |
| #8mars, #2030 | Temporal expression in hashtags (online only) |

## 4. Results

The research question of this study was: how to identify stakeholders' living contexts in SPPs and what patterns do stakeholders implicitly use to represent this context in SPP inputs? To answer this research question, we focused on semantic, spatial, and temporal patterns that stakeholders use to share some properties of their living contexts. As depicted in Table 1 and Figure 2 we identified 26 patterns following our semantic, spatial, and temporal analysis of SPPs comments (see Table A1 in Appendix A, Tables A2–A6 in Appendix B). As the collected data is from both online and offline participation processes, we observed that some patterns were specific to online participation while all the other patterns were independent of the means used to participate. In the next subsections, we present and explain the final patterns identified for each dimension.

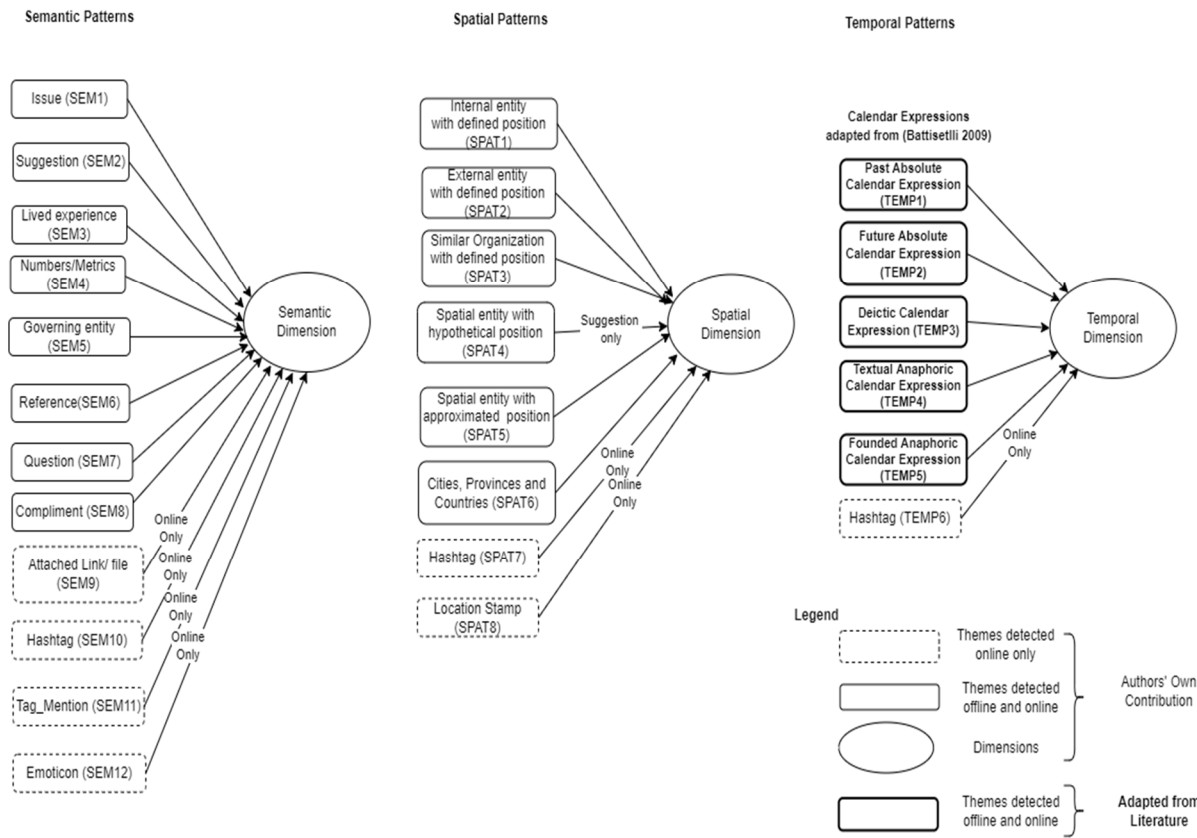

**Figure 2.** Semantic, spatial, and temporal patterns in SPP data.

## 4.1. Semantic Patterns in SPP Data

Our qualitative analysis of SPPs data allowed us to identify twelve semantic patterns that were: "issues (SEM1)", "suggestions (SEM2)", "lived experiences (SEM3)", "numbers/metrics" (SEM4), "governing entities (SEM5)", "references (SEM6)", "questions (SEM7)", "compliments (SEM8)", "attached link/file(SEM9)", "hashtag (SEM10)", "tag_mention (SEM11)" and "emoticon (SEM12)". For instance, a participation comment could contain only one semantic pattern, or combine several semantic patterns. For example, the following comment from case 1 has one semantic pattern, a "suggestion (SEM2)": "I am making a proposal that ethics and sustainable development courses be more widely taught in engineering programs" (case 1). However the following comment combines two semantic patterns: "issue (SEM1)" and "Number/metrics (SEM4)":"A park bench with a piano that costs $20,000, an amount close to the average salary of a citizen, it's quite extravagant and bourgeois" (case 2).

As shown in Tables A2 and A3 (Appendix B), we found twelve semantic patterns that participants use to semantically represent their living contexts. We observed that some semantic patterns were used more frequently than others. These patterns are "Issue (SEM1)" that was found in more than 52% of the comments, "suggestion (SEM2)" in more than 38% of the comments and "Lived experience (SEM3)" in more than 19% of the analyzed comments.

We observed that when stakeholders express issues (SEM1), they generally use negative expressions such as for example "the obstacles we see", "it worries me that", "there is no initiative for", "which poses a problem" (cases 2 and 3). An issue is something that the stakeholders are aware of and that is specific to their specific environments. On the other hand, we observed that when stakeholders made suggestions, they usually used expressions such as "I suggest, I make the proposal to, Let's do etc." (cases 1 and 2). A suggestion (SEM2) is an idea or a plan to be considered. It implies a certain fact or situation that a stakeholder wishes to achieve. A suggestion can be brief or developed through

arguments that are often important to consider because they reflect some properties of the stakeholders' living contexts. In the following, we present two suggestions in the same topic with different levels of specificity (brief and developed):

"I suggest building a place conducive to gatherings!" (Case 2) or "With a local population involved and interested in its neighborhood, it would be important for [organisation case 2] to offer its citizens a multifunctional public space that resembles them. I make the proposal to put in place a unifying public place for people from the neighborhood or elsewhere." (Case 2). Finally, stakeholders referred to stories or lived experience (SEM3) that relates to the topic they discuss to argue the relevance of their opinions. Participants use specific expressions to share a lived experience such as: "I experienced this", "for having lived it" (Cases 1 and 4). Besides the three most cited patterns, we present hereafter the other patterns:

- Participants can share "numbers/metrics (SEM4)" to quantitatively argue their opinion. Numbers/metrics are used often to show a critical situation (e.g., "Tunisians throw away a billion plastic bags annually. It is an ecological disaster in good and due form. The other 700,000,000 bags are distributed by various other economic operators including municipal and central markets") or to make a specific suggestion (e.g., "Let's reduce the maximum speed to 30 km/h" (Case 1).
- Participants can also share "References (SEM6)" or "governing entities (SEM5)". A "reference" is a pattern that is defined as an "article, initiative, author, celebrity, public figure or a program that is evoked in a participation comment and to which one can refer either by a name or by an abbreviation". "Governing entities" are incorporated or unincorporated association, committees, persons, or any other entity that has authority to which stakeholders refer in their comments.
- Stakeholders can also ask questions ("Questions (SEM7)") to acquire knowledge about their living context or to make suggestions. Participants can make compliments ("compliments (SEM8)") to express their satisfaction about decisions or actions taken by their decision-makers in their living context.

Finally, from all the identified patterns, we found that there are four semantic patterns that were specific to online comments. These patterns were "Attached link/file(SEM9)", "Tag_mention(SEM11)", a "hashtag (SEM10)" with a meaningful insight or an "emoticon (SEM12)" to express an emotion. These patterns were considered as semantic as their use enhances the meaning of the comment and could contribute to understand the living contexts. Links or files can contain relevant information related to the topic discussed in the SPPs comment and to the living contexts of stakeholders. "Tag_mentions" and "hashtags" are generally used in participation through social media channels [43]. A tag_mention is a label to engage an individual, organization, or any entity with a social profile when they mention them in a post or a comment [43]. So, a tag in a SPPs comment, refers to an individual or to an organization that stakeholders consider as relevant in their living contexts. A hashtag is a feature provided by social media channels enabling to highlight keywords of topics within a comment [43]. An emoticon is a symbolic expression that stakeholders use to symbolize a facial expression, an emotion, or an attitude [44]. It is a small icon composed of punctuation characters.

*4.2. Spatial Patterns in SPP Data*

As shown in Tables A4 and A5 in Appendix B, we identified eight spatial patterns that participants use implicitly to identify their living contexts: "internal spatial entity with a defined position (SPAT1)", "external spatial entity with a defined position (SPAT2)", "similar spatial entities with defined positions (SPAT3)", "Internal spatial entity with a hypothetical position (SPAT4)", "spatial entity with an approximated position (SPAT5)", "Cities, Provinces and countries (SPAT6)", "spatial entity in a hashtag (SPAT7)", and "spatial entity through a location stamp (SPAT8)". We observed that some spatial patterns were used more frequently than others. These patterns are "Internal entity with defined position (SPAT1)" that was found in more than 46% of the comments, "External entity with defined

position (SPAT2)" in more than 15% of the comments and "Spatial entity with approximated position" (SPAT5)" in more than 14% of the comments.

As stated in the theoretical background section, spatial pattern, on which we are focusing in this research, are places or entities fulfilling a localization function [33]. Spatial entities (SE) with defined position (DP) are places with a specific location [27]. They are entities fulfilling a localization function (occupy a position) [27]. These entities can be determined through their contours by means of coordinates [33] where stakeholders give a very precise indication about the place. We found three spatial entities with defined position: "internal spatial entity with a defined position (SPAT1)", "external spatial entity with a defined position (SPAT2)", and "similar organization with a defined position (SPAT3)". These patterns were categorized according to the "frame of reference" of each organization of each case as internal, external or similar to each organization. For example, for SPAT1, if the organization is a city, then internal SE with DP are all the spatial entities within the frame of reference of the city such as districts, parks, or streets (see examples Tables A4 and A5 Appendix B). The external SE with a DP (SPAT2) are anchored outside the frame of reference of the organization. They are all spatial entities with a defined position that do not belong to the frame of reference of the organization. As examples for a city, we mention parks or districts that are outside the city. Finally, a similar SE with a DP (SPAT3) is of the same type as the organization interested in the SPPs; if the organization is a city than a SPAT3 would be another city. Our analysis indicated that generally stakeholders referred to similar organizations to make a comparison or to give an example or to propose a project.

Internal SE with a hypothetical position (SPAT4) are spatial entities that do not exist but stakeholders indicate a specific position that they might occupy in the future. For example, a stakeholder stated: "I suggest setting up a public square at the corner of Canardière and 4th avenue" (Case 2). In this case, the public square does not exist but the streets exist.

The SE with an approximated position (SPAT5) are places to which stakeholders do not give very specific indication about the location. For example, "the organic community garden" in the following example is an SE with approximated position: "These fertilizers can be used in the organic community garden" (Case 1).

In fact, there are many organic community gardens inside and outside the organization of (case 1) which requires further examination to locate the garden that the stakeholder was pointing to. Stakeholders also refer to cities, provinces, and countries through their participation. These three types of spatial entities are grouped into the category of "Cities, Provinces, and countries (SPAT6)". Eventually if the organization concerned with the SPP is a city, other cities would be "Similar cities SPAT3 instead of SPAT6".

Finally, we found two spatial patterns specific to online participation that are "spatial entity in a hashtag (SPAT7)", and "spatial entity through a location stamp (SPAT8)". In fact, stakeholders might use patterns such as "location stamp" to identify a location or "a hashtag" to refer to a specific location. These two online patterns were enabled through features provided by social media channels to share spatial locations or coordinates. Indeed, it is important to underline that these two online patterns can contain the same spatial information that we can find in other spatial patterns, specifically those with defined position (e.g., city, internal SE with DP etc.). The only difference was in the way of representing the information through the features provided by the used participation tools.

*4.3. Temporal Patterns in SPP Data*

Following our analysis of SPPs data [29], we identified six temporal patterns: "future absolute CE (TEMP1)" "Past absolute CE (TEMP2)", "deictic CE (TEMP3)", "textual anaphoric CE (TEMP4)", "founded anaphoric CE (TEMP5)", and "temporal hashtag (TEMP6)" (see Tables A6 and A7 in Appendix B). We observed that some temporal patterns were used more frequently than others. These patterns were "Deictic calendar expression (TEMP3)" that were found in more than 19% of the analyzed comments and "Past absolute calendar expression (TEMP1)" in more than 12% in the comments. As emphasized in

Section 2.3, five out of the six temporal patterns that we detected in SP data were similar to the four main types of calendar expressions presented in [28] which were: Absolute CE, Deictic CE, Textual anaphoric CE and Founded anaphoric CE. In Figure A1 (Appendix C), we explain how we adapted the model to emerge temporal patterns that were specific to the identification of patterns of the living contexts in SPP data.

Thus, in our final model of temporal pattern we augmented the absolute CE by two patterns "past absolute" and "future absolute" and we added "the temporal hashtag" pattern. Hereafter, we explain each of the temporal patterns. Future absolute CE is an important information in SPPs as decision-making processes may be concerned with projects or policies to be implemented in the future. For example, we have the following comment: "The [organization case 1] would offer the passes at a lower cost and, since it is about sustainable development, [organization case 1] could write off that expense in this fund? Could this offer a trial for a year starting in September 2018?" (Case 1). In this comment, "year starting in September 2018" is a future absolute CE. Either with specific suggestions about projects to be implemented in future, or by highlighting relevant predictions/forecasts related to the subject discussed, future absolute CE reflect the expectation of stakeholders about their living contexts in the future.

"Past absolute CE" refers to an absolute date or a duration in the past. This temporal information is significant as stakeholders may try to point to a specific period in the past where for example issues, decisions or projects have taken place. Usually, these issues, decisions or projects are worth knowing to understand the living contexts. Let us consider the following comment: " ... *A savage deforestation that has lasted since 2011 in the total indifference of the forest services to clear land and concrete it to the maximum despite the law and common sense, forever destroying ecosystems to cover with dust of cement, cypress, thyme and rosemary, wonderful flora with which nature has endowed what was a haven of peace. Let us affirm our solidarity and show our support to those like Nawaat who were part of the quest for the truth about the abuses which destroy all that we have most precious, our nature, our natural environment source of wealth and oxygen*." (Case 4). In this case, "since 2011" is a "past absolute CE".

Deictic CE are temporal CE requiring to know the date when a SPP comment was drafted. For example, a stakeholder asking the following question: "We don't even count the number of outages on the orange line since the start of the year @stminfo, compensation for subscribers?" (Case 3). In this case, it becomes important to know the date when the commentary has been posted so that the organization can determine if actions need to be taken. A Deictic CE could be either a past absolute CE (which is the case in this example) or a future absolute CE.

A textual anaphoric CE pattern is used in comments telling a story and highlighting a succession of events where stakeholders for example share a living experience. Their identification depends on an antecedent calendar expression that is identified earlier in the text (e.g., a SPP comment). Just like deictic CE, they could be converted to past or future absolute CE and bring a similar added value in terms of temporal awareness about the living contexts. Let us consider the following example where the textual anaphoric CE (3 years after) and its antecedent CE (in 2013) are outlined: "*there is an initiative that was launched in 2013 at the time, among other things, of the rector and the leaders of the health establishments in the region which aimed to tackle so that promote partnerships with the communities have proposals unique so the idea was to say how can we be interested in research in health and social services other than by the strict end of the cure, or, of the molecule or, of the solution to a particular problem so uh It's not easy to broaden perspectives, but we were able to do so by organizing forums like this one, which brought together, 150 or 180 key people in health and social services research, health service delivery and social services as well, from the world of private research, then from the world of private companies involved in manufacturing. This process led to a common thread around which we should articulate our research efforts, namely the concept of sustainable health. 3 years after the start of this initiative ... my deep conviction is that it is possible to break the sylos*". (Verbatim from Case 1).

Founded anaphoric CEs are based on the knowledge of the world. They can correspond to a specific "date" or "duration" but also to a more or less "fuzzy" date. For instance, a participant in case 1 refers to a founded anaphoric CE to warn the risk of making decisions that date back to a certain outdated time: « . . . we shouldn't do a bit like in the 1950s with programs aimed at women and others aimed at men . . . » (case 1).

Finally, temporal hashtags are specific to online participation tools. As for the semantic and the spatial dimensions, stakeholders use features provided by new technologies to share their temporal perception about their living context. In temporal hashtags, stakeholders can share any of the previous six temporal patterns identified in SPP inputs.

### 4.4. Research Outcomes

Thus far, we presented 12 semantic patterns, 8 spatial patterns and 6 temporal patterns to identify the living context of stakeholders in SPPs data. Following the axial analysis, continued questioning of our data led as to note relationships between semantic, spatial, and temporal dimensions. Moreover, as we detected patterns that were specific to online participation (online only patterns) for the three dimensions, we note that information technologies could play an important role in highlighting the stakeholders living contexts in SPPs.

### 4.5. Relationships between Semantic, Spatial and Temporal Dimensions

The first relationship between dimensions is complementarity. In our theoretical framework, we conceptualized semantic, spatial, and temporal dimensions as three separate dimensions of the stakeholders' living contexts. Based on our interpretation of data, we came to understand that spatial and temporal patterns are used by stakeholders to complement the semantic patterns they provide. Therefore, spatial and temporal dimensions are complementary to the semantic dimension.

In fact, to give sense to their living contexts in their comments, stakeholders use semantic patterns. To be more specific about the information provided through the semantic patterns, they may provide spatial (e.g., spatial entities with defined position) and/or temporal information (e.g., future absolute calendar expressions). Indeed, 100% of the SPP comments analyzed had a semantic dimension providing at least one or more semantic patterns. More than 83% of these comments had at least a spatial pattern and 36% of these comments had at least a temporal pattern. However, 0% of the analyzed SPP comments had a spatial and/or a temporal dimension without providing a semantic pattern. Based on our findings, we note that the identification of spatial and temporal patterns in SPP data must be directly related to semantic patterns. In other words, the identification of spatial entities (e.g., cities, similar organizations) and temporal expressions (e.g., future calendar expression, past calendar expression) in SPPs data without relating to semantic entities (e.g., issues, suggestions, compliments), would not bring an added value in understanding the living contexts of stakeholders from SPP data.

Second, we discovered that correlations can be detected between patterns of different dimensions (see Table A8 in Appendix D). A correlation between patterns means a connection between two or more patterns in a way that they occur together in a repeated manner in comments. Again, detected correlations are between the semantic dimension and other dimensions. Thus, correlation could be considered as a sub-relationship of complementarity.

For example, in some comments, we noted the co-occurrence of the following patterns in different comments: "Suggestion (SEM2)" and "Spatial entity with hypothetical position (SPAT4)", and "suggestion (SEM2)" and "Future absolute Calendar expression (TEMP2)". These correlations depict that in some cases, stakeholders who provide suggestions (SEM2), provided also hypothetical (SPAT4) locations or future dates in relation to the suggestion (TEMP2). For instance, the pattern "Future absolute calendar expression (TEMP2)" was found in 2% of the analyzed comments. In 60% of these comments, (SPAT4) was used with a suggestion (SEM2). In 40% of these comments, (TEMP2) was used with a suggestion

(SEM2). As the frequency of future calendar expressions (TEMP2) was not very high in our data sample, the observation on the possibility of detecting such correlations between patterns was noted but could be further validated in future research. In Appendix D, we present comments from different cases to emphasize the correlations detected between these patterns.

Based on our interpretation of these relationships, we note that the identification of spatial and temporal information in SPP data will be needed to help decision-makers complement their understanding about the living contexts. To fulfill this need, spatial and temporal dimensions should be identified and analyzed as complementary dimensions to the semantic dimension of the stakeholders' living contexts. This way, spatial and temporal dimensions should help to locate in space and time, the semantic information (e.g., issues, suggestions, lived experiences) that stakeholders provide in SPPs (input).

### 4.6. The Role of Information Technology in Highlighting Stakeholders' Living Contexts

As emphasized in results, "online only" patterns were detected for each dimension. In fact, online only patterns in SPP data were resulting from the use of information technology (IT)-based participation tools. These participation tools provide several IT-features to users such as the possibility to attach a file in participation platforms, and the hashtag (#) and the tag (@) in social media platforms. Our interpretation of online only patterns included a comparison between online only and other patterns.

Our main observation was that an online only pattern can provide the same information as an offline pattern. It was only the way the information was presented in the data that differed (e.g., adding a hashtag, stamping a location instead of a simple text). Moreover, we remark that "online only" patterns were more observable or explicit than other patterns in SPP data as they were preceded by symbols or special characters.

For example, as emphasized in results, using a hashtag (online only pattern), a stakeholder could either provide a semantic information: e.g., an "issue (SEM1)", a temporal information e.g., "a future absolute calendar expression (TEMP2)" or a spatial information e.g., a "city (SPAT6)". By definition, hashtags aim to highlight keywords or topics within a text, so, thereby making them more explicit in data.

Another example, is the "tag_mention (SEM11)" pattern. In SPPs, some stakeholders used this IT-feature to mention either a "reference (SEM6)" or a "governing entity (SEM5). However, when the participation was physical or when the tool used did not provide the "tag_mention" feature (for example "participation platform (case 2)"), we remarked that stakeholders just mentioned the "reference" or the "governing entity" in textual manner, which made it less observable or explicit (or more implicit) in SPP data.

Similarly, for "Location Stamp (SPAT8)", which is an online pattern that is provided by social media tools in our case studies. Location stamps enable to highlight in a more explicit manner spatial patterns that are often implicit in SP data (such as "Internal SE with DP (SPAT1)" and "External SE with DP (SPAT2)". According to our analysis, we note that despite the abundance of spatial patterns in SP data, the use of "location stamps" by stakeholders was very limited. Even when the participation tool allowed to use a location stamp (e.g., social media channel), stakeholders textually mentioned places they wished to highlight, which made their detection more difficult in textual data.

The analysis of online patterns vs offline-online patterns enabled us to note the importance of providing suitable IT-features that were sensitive to the semantic, spatial, and temporal dimensions. For example, specific IT-features that enable to identify issues, spatial entities and temporal expressions would help stakeholders to more explicitly highlight and communicate important information about their living contexts to decision-makers in SPPs. Indeed, IT-features that are mainly provided by social networks should be extended to all other e-participation means. Based on our interpretation of the data, we note that IT could help to build the necessary capacities to automatically identify the stakeholders' living contexts in future SPP tools.

Hereafter, we present a conceptual model for the stakeholders' living contexts identification in SPPs. In this model, we suggest that the interrelated semantic, spatial, and temporal dimensions, as well as IT, are central in highlighting the living contexts in SPPs. Moreover, we show the importance for decision-makers to capture and to understand the semantic, spatial, and temporal patterns in SPP data to ensure a decision-making that is consistent with and responsive to stakeholders' living contexts.

### 4.7. A Conceptual Model for the Stakeholders' Living Contexts Identification in SPPs

For SPPs to be responsive to stakeholder input, an understanding of their living contexts will be needed. As illustrated in Figure 3, stakeholders provide inputs in SPPs with the aim to influence decision-making processes about significant choices affecting their communities, and consequently, to lead to better decisions [2]. The SPPs generate data that decision-makers have the challenge to analyze and to understand in order to help stakeholders achieve their communities' objectives [4,5]. The concept of the stakeholders' living context, as illustrated in Figure 3, is at the core of SPP data. Drawing on our findings, we conceptualized the stakeholders' living context in SPP data as an adaptive collection of semantic, spatial and temporal patterns that were interrelated and whose identification and understanding holistically support decision-making, thereby enabling SPPs outcomes to be grounded in analyzing the context closely [6]. Within the process of identification and understanding of the living contexts, existent, and emerging technologies relevant to participation such as social media, participation platforms, among others, will play an important role in supporting SPPs data collection, analysis, and representation [40].

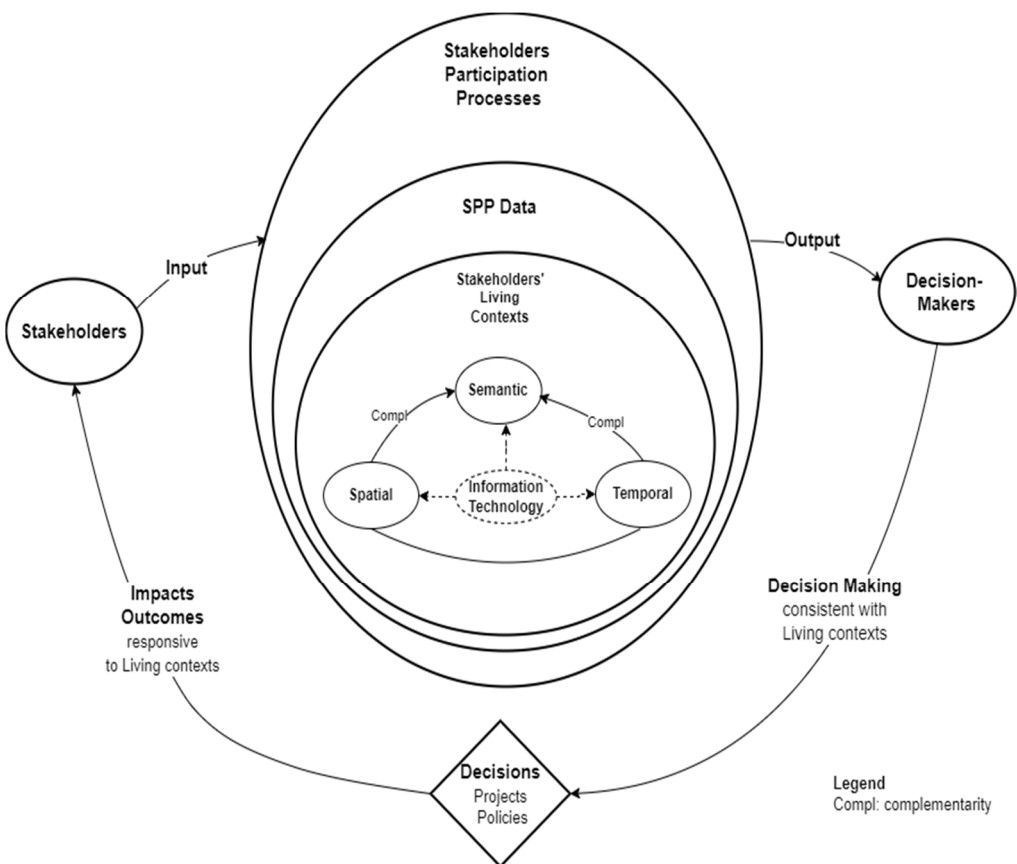

**Figure 3.** A conceptual model for the stakeholders' living contexts identification in SPPs.

To further elaborate the role of the identification and the understanding of stakeholders' living contexts for generating effective SPPs outcomes, we emphasize the interactions that are needed between stakeholders and decision-makers through SPPs.

As described previously, SPPs represent the space in which stakeholders present their objectives and priorities that are embedded on their living contexts. As such, they require the collection of a broad range of data from stakeholders with a diversity of views and needs. Equipped with these data (output from SPPs), decision-makers can engage in a process of analysis and discussions of semantic, spatial and temporal patterns, which allows for more informed decisions, better definitions around projects and policies and the identification of the critical challenges for responding to stakeholders' living contexts. Semantic, spatial, and temporal patterns provide not only a rich information to inform decision-makers but also support their capacity to build feedback about how decisions respond to and impact stakeholders living contexts over time [45].

Here, information technology has an important role to play in supporting the identification of living contexts in SPPs. As emphasized previously, participation tools that are endowed with IT-features could considerably help in the process of identification and analysis of semantic, spatial, and temporal patterns in SPP data. However, the use of IT is not an end in itself, but a means to achieve traditional participation goals [8]. Existing and emergent technologies help to support the process of identification, analysis, and representation of the patterns in data of living contexts. The concept of the living contexts within SPPs data remains central to SPPs whether stakeholders use IT-based or traditional participation tools. The challenge is in capturing and analyzing patterns of the living contexts form data, and IT has the potential to permit automatic capture, deep and detailed analysis, and intuitive representation of these patterns [40]. This interpretation explains why IT is in dotted lines in our proposed model.

Based on our findings, we suggest that, to help stakeholders explicitly identify their living contexts and for decision-makers to better capture the living context, IT-based tools that are sensitive to the detection of semantic, spatial, and temporal patterns will be needed. Moreover, better effort should be incurred to encourage stakeholders make-sense and appropriate IT-features in electronic tools [44]. Using innovative technologies, such as participatory GIS [40], spatial and temporal patterns that are complementary to understanding the semantic patterns in SPPs data, would be valued. In addition, we suggest that other technologies including data analytics coupled with spatio-temporal visualization could allow for automatic identification of complementarities and correlations between patterns. Such technologies can support stakeholders in explicitly identifying their living contexts and help decision-makers to better capture it, understand it and visualize it to ensure consistent and responsive decision making.

To ensure effective SPPs, purposeful interactions must occur between stakeholders and decisions makers. The identification and the understanding of the semantic, spatial, and temporal dimensions of the stakeholders' living contexts are at the core of these interactions. Through new context-sensitive technological capacities, organizations concerned with SPPs can make concrete progress toward building effective SPPs leading to smarter projects and policies implementation [45].

## 5. Discussion

With the goal to understand how stakeholders identify their living contexts in SP comments, we analyzed in this study data from four cases studies, and we proposed an empirical set of semantic, spatial, and temporal patterns. Following a qualitative approach, 26 final patterns emerged that we classified into 12 semantic patterns, 8 spatial patterns and 6 temporal patterns. Moreover, the relationships between dimensions as well as the role of information technology in highlighting these dimensions were emphasized. Drawing on these finding, a conceptual model of the stakeholders' living contexts identification in SPPs was proposed, presenting practical implications of our findings. The following theoretical implications arise from this study.

First, the semantic patterns identified in the empirical model were complementary to previous research. Previous semantic analysis in literature enables to categorize comments according to general topics based on "words" detection and classification [25,26,46].

Our semantic analysis of SP data used "expressions" rather than "words" and proposed 12 semantic patterns that were complementary to topics' detection. Indeed, in addition to topics, we note that stakeholders identify "issues", "suggestions", "lived experiences", "governing entities" etc. that could be detected in SP comments to understand their living contexts. By detecting these semantic patterns, decision-makers can develop a better understanding of the context in which their stakeholders live, leading to smarter and more informed decisions. Indeed, the semantic patterns identified in this research demonstrate that the intelligence of organizations and cities could be enhanced through stakeholder identification of the different semantic patterns, to which spatial and temporal patterns bring a complementary view.

Second, we note that stakeholders identify "purely spatial entities" [33] in SP comments to refer to their living contexts. We identified eight spatial patterns that stakeholders use to identify their living contexts in SP comments. The identification of spatial patterns in SP data requires determining the frame of reference which consists of the geographic location of the organization concerned with the SPP data [27,31]. The frame of reference of the organization enables to locate the spatial entities detected in SP data and to endow them with spatial relationships that characterize their relative fixity during a given period. Our findings were consistent with previous research that stipulated those frames of reference are fundamental to locate and to follow the evolution over time of spatial entities, to which, in our study, stakeholders referred while they identified their living contexts in SP data [33]. However, to the best of our knowledge, no previous research has analyzed the spatial dimension and identified spatial patterns that stakeholders use in their participation comments to share their sense of place. We believe that detecting spatial dimension, in combination and complementarity to semantic patterns will help decision-makers to develop their knowledge about stakeholder relationship to places and sense of place [30].

Third, about the temporal analysis, our focus in this research was on the identification of temporal patterns that stakeholders use in SP data to temporally identify their living context. For that, we identified six temporal patterns that stakeholders use in SP comments to enable their anchoring in a time axis (or a calendar system). Our analysis aligned with the first level of temporal analysis highlighted by the authors in [29], which consists of the identification of temporal expressions in texts. Nonetheless, the second level which consists of "calculating the temporal ordering of events in a text" [29] could be considered as a perspective for future research.

Moreover, as emphasized in the theoretical background, four temporal orders (modal, aspectual, enunciative and temporal) for the detection of time in texts are identified in literature [29]. Following our temporal analysis of SP data, we note that the identification of temporal expressions in SP data are consistent with the "temporal order". The temporal order asks if the temporal content is located in the present, past or future. The modal orders ask about the certainty of the content, which does not apply in SP since SP comments should be analyzed in an objective manner. The aspectual order considers the aspectual properties of the lexical level (verbs, nouns, objectives) and grammatical markers, which is out of the scope of our research. Finally, the enunciative order considers several interlocutors in text units, which does not apply in our research as participation comments are individual. The findings of this study establish a fundamental ground to the identification of the stakeholders' living context from SPP data. The empirical semantic, spatial, and temporal patterns as well as the conceptual model of the stakeholders' living contexts identification presented so far offer a better comprehension of the benefit that SPP data might outstand in understanding the living contexts of stakeholders.

Our findings have implications for both traditional and electronic participation as SP data is generated using both means. For traditional participation, qualitative analysis could be applied to detect semantic, spatial, and temporal patterns from collected data

For e-participation, tools that are sensitive to the stakeholders' living context, e.g., providing IT-features to explicitly identify patterns, should also be developed in future.

Moreover, our empirical model of patterns can be extended according to future needs. As there has been a growing interest in tools and methods based on the notion of space and place, in the last few years, such as volunteered geographic information (VGI) [47] and softGIS methods [48–50], the patterns identified in this research could be used to help decision-makers to develop a better qualitative understanding of social synergies in cities and organizations.

Overall, our research aligns with recent studies demonstrating that the intelligence of an organization (e.g., a city) is related to its stakeholders' (e.g., citizens) ability to understand and to share events or phenomena that characterize its internal dynamics and external relations [30]. Through the semantic, spatial, and temporal patterns identified and classified in this research, this demonstration is now concrete and future research and tools may be based on the patterns to develop features to better capture the events and phenomena that stakeholders are living, and to enhance the intelligence and the responsiveness of decision-makers to stakeholders' living contexts [47–49].

## 6. Conclusions

In this research, we propose a model of semantic, spatial, and temporal patterns for the identification of stakeholders' living contexts in SPPs data. Moreover, we present a conceptual model where we emphasize the relationship between the three dimensions of patterns, the role of IT and the importance for decision-makers to capture these patterns to enhance their responsiveness to their stakeholders' living contexts.

Notwithstanding its promising findings, this study has some limitations. The first limitation consists of not considering data from all participation tools such as emerging participative technologies e.g., Volunteered Geographic Information (VGI) and 3D sophisticated visualization platforms. The choice of the participation tools in this study was guided by the nature of the targeted data, which were mainly textual and which were generated in a natural way. As sophisticated participation tools are already endowed with spatial and temporal functionalities and IT features, these tools may push users to make sense and to appropriate these features and to probably generate patterns that are different from the patterns which are generated spontaneously in a simple textual way (e.g., maps, visualization features). For this reason, we have omitted to refer to these kind of tools as our objective was to understand the way the living context is naturally expressed by stakeholders in SP data. However, our results confirm the relevance of GIS-based tools and provide important knowledge to consider in the design and implementation of these tools in the future.

Several avenues for future research arise from this study. First, future research could investigate the possibility to adapt existing artificial intelligence algorithms to automatically apply the semantic, spatial, and temporal contextualization approach through automatic identification of patterns in SP data. According to Gartner's report on emerging technologies, http://www.gartner.com/document/3383817?ref=solrAll&refval=175496307&qid=34ddf525422cc7 (accessed: 25 May 2016). incorporating machine learning, in particular, enhances the decision-making process and provides valuable insights from large-scale data. Detecting semantic, spatial, and temporal patterns through machine learning techniques could help capturing the living contexts form SP data and thus helping decision-makers make more effective decisions generating better outcomes and impacts. Thus, the finding of our research offers theoretical background for future participative technologies using artificial intelligence techniques. Second, future studies could be based on our finding that are derived from "naturally occurring SP data" analysis to evaluate emerging participative technologies such as VGI and 3D sophisticated visualization platforms and to determine how much these tools are representative of stakeholders living contexts based on semantic, spatial, and temporal patterns [45]. Third, future research could confirm frequency of patterns depending on the nature of the tool used (online, offline, social media, participation platform etc.) based on a larger amount of SP data. Knowing the frequency of patterns according to the participation tool would be helpful to identify relevant patterns for each

participation tool. Finally, future research could focus on the detection of correlations between patterns with the aim to detect two or three-dimensional level patterns, depending on the patterns present in the SP data.

**Author Contributions:** Conceptualization, A.M., S.M. and S.D.; methodology, A.M.; software, A.M.; validation, A.M., S.M. and S.D.; formal analysis, A.M.; investigation, A.M.; resources, A.M.; data curation, A.M.; writing—original draft preparation, A.M.; writing—review and editing, A.M., S.M. and S.D.; visualization, A.M.; supervision, S.M. and S.D.; project administration, A.M. All authors have read and agreed to the published version of the manuscript.

**Funding:** This research received no external funding.

**Informed Consent Statement:** Not applicable.

**Conflicts of Interest:** The authors declare no conflict of interest.

## Appendix A

**Table A1.** Case studies description.

| Case Number | Case 1 | Case 2 | Case 3 | Case 4 |
|---|---|---|---|---|
| **Organization type** | University | District | Public collective transport company | City |
| **Size** | 45,000 students 7050 employees (including professors, teachers, other employees) | 107,885 residents | 32,760 customers | 2,426,000 habitants |
| **Country/city** | Canada/Québec | Canada/Québec | Canada/Montréal | Tunisia/Tunis |
| **Nature of participation process** | Participatory campaign for strategic planning | Citizen participation process concerning the construction of a public square | Customers participation about the company's services—Tweets | Citizen participation about their city—A FB page |
| **Period in which comments were made** | Between November 2017 and February 2018 | Between 2015–2016 | Between March 2017 and April 2017 | Between January 2017 and December 2017 |
| **Nature of the participation process** | Solicited | Solicited | Spontaneous | Spontaneous |
| **Data collected** | -Forums -Web form | -Participation dedicated platform | -Social media (Twitter) | -Social media (Facebook) |
| **Overage size of a comment (by word)** | -Forum: 273 words -Web Form: 171 words | 33 words | 25 words | 63 words |
| **Number of SP comments generated** | Collected: -Forums: 156 (42,540 word) -Web Form: 297 (50,740 word) Analyzed: -Forums: 33 -Web Form: 29 Total: 62 | Collected: 126 (4185 word) Analyzed: 70 | Collected: 3587 Tweets (89,600 word) Analyzed: 68 | Collected: 791 (49,941 word) Analyzed: 44 |
| **Collection techniques** | -Recording -CSV file (provided by the university) | -CSV file (provided by the city) | -CSV file (Collected through Twitter API) | -CSV file (Collected through Facebook API) |
| **Language** | French | French | French | French |

## Appendix B

**Table A2.** Semantic Patterns in both online and offline SP data.

| Semantic Pattern | Code Name | Code Meaning | Illustrative Quotes |
|---|---|---|---|
| **Issue** | SEM1 | An important topic or problem for debate or discussion | "For student parents, the reality is often one of reconciling family, studies, work. The difficulties they encounter are varied, touching on scheduling conflicts, poverty, exhaustion, problems accessing child care" (case 1)<br>"AGAIN! Metro: long outage of more than an hour on the orange line" (Case 3)<br>"For your information, Tunisians throw away a billion plastic bags annually. It is an ecological disaster in its due form." (Case 4) |
| **Suggestion** | SEM2 | An idea or plan put forward for consideration | "I am making a proposal that ethics and sustainable development courses be more widely taught in engineering programs" (Case 1)<br>"I suggest building a public square at the corner of Canardière and 4th avenue. It is already a public square but it is not frequented due to the lack of attraction" (Case 2) |
| **Lived experience** | SEM3 | A representation and understanding of human's experiences, choices, and options | "We've been stuck for an hour and a quarter and our children are waiting" (Case 3)<br>"Last summer, when we had a piano on 3rd Avenue (which I would love to see again this year too!), We could see people of all generations meeting there." (Case 2) |
| **Numbers/metrics** | SEM4 | A number or a measure of something. | "Let's reduce the maximum speed to <u>30 km/h</u>" (Case 1)<br>"Why is the service so slow to Côte-Vertu? <u>25 min</u> to do <u>3 stations</u>, and again stopped." (Case 3)<br>"A park bench with a piano that costs <u>$20,000</u>, an amount close to the average salary of a citizen, it's quite extravagant and bourgeois" (Case 2) |
| **Governing entity** | SEM5 | An incorporated or unincorporated association, committee, person, or any other entity that has authority | "Have in its database all professionals and executives of<br><u>the Ministry of International Relations</u>" (Case 1)<br>"for compliance with the specifications of<br><u>the Ministry of Women and Family Affairs</u>" (Case 4)<br>"To restore confidence in the<br><u>SPVM (Montreal Police Department)</u>, transparency is needed on a permanent basis, not periodically" (Case 3) |
| **Reference** | SEM6 | Article, initiative, author, celebrity, public figure, program listed in participants comments and to which on can refer either by a name or an abbreviation. | "to be inspired by the multiple proposals and ideas that have emerged as part of<br><u>the Idex excellence initiative</u> in France." (Case 1)<br>"The STL offers a compensation program to dissatisfied customers" (Case 3)<br>"The UN organized the<br><u>World Road Safety Film Festival</u>" (Case 4) |
| **Question** | SEM7 | A sentence worded or expressed so as to elicit information. It generally refers to an issue or a suggestion or both in the participation context. | "When will @ amt_info finally ban smoking on the docks?" (Case 3)<br>"what to do in case of fire? And the most shocking question how this promoter obtained his authorization from the civil protection?" (Case 4) |



**Table A2.** *Cont.*

| Semantic Pattern | Code Name | Code Meaning | Illustrative Quotes |
|---|---|---|---|
| **Compliment** | SEM8 | A polite expression of praise or admiration. | "I want to thank you for these user-friendly, innovative and ecological improvements. With this development, you enhance the look and quality of your infrastructures" (Case 1) <br> "Bravo to the driver of the Express 550 who kept her smile and was very patient during the traffic jam earlier!!" (Case 3) <br> "Thank you once again to the authorities for their responsiveness, and thank you to the members of the group who shared or reacted to the post"(Case 4) |

**Table A3.** Semantic Patterns in SP data online only.

| Online Only Semantic Pattern | Code Name | Code Meaning | Illustrative Quotes |
|---|---|---|---|
| **Link or document/file** | SEM9 | Attached to the comment. Online only. | "These two challenges, as reflected in the proposed mandate for the future, included in the document "attached" to this commentary" (Case 1) <br> "It looked like that at the Berri-UQAM metro station a few minutes ago before the announced resumption of service on the orange line https://t.co/jA93Oeba1x (accessed on 4 April 2022)" (Case 3) |
| **Tag** | SEM11 | A label attached to someone or something for the purpose of identification or to give other information. | "The phone thief at Jarry station who runs away from a girl and a man @stminfo @SPVM" (Case 3) <br> "@stminfo when will the Azurs be on the green line and low fares for low-income people? #polmtl #Transport @CraigSauve" (Case 3) |
| **Hashtag** | SEM10 | A word or phrase preceded by a hash sign (#), used on social media websites and applications, especially Twitter, to identify digital content on a specific topic. | "Escalator that has gone down for 1 week. Today all the stairs are broken! # DuCollège # accessibility" (Case 2) <br> "every time there is a #delivery the #customers #tunisians #corrupted ask for 200 dinars #Share please #douane #tunisienne #corruption"(Case 4) |
| **Emoticon** | SEM12 | A representation of a facial expression such as:-) (representing a smile), formed by various combinations of keyboard characters and used to convey the writer's feelings or intended tone. | "ok problem on the Orange line is it possible to stop the messages after 30 s 😵" (Case 2) <br> "This is what we call, walking on eggshells, we saw nothing and the cars coming in the opposite direction dazzled us 😬😬" (Case 4) |

**Table A4.** Spatial patterns in both online and offline SP data.

| Spatial Pattern | Code Name | Code Meaning | Illustrative Quotes |
|---|---|---|---|
| **Internal spatial entity_ Defined position** | SPAT1 | A spatial entity with specific location that is internal to the organization concerned with the participation data. | "Despite the presence of parks, there is a lack of public space in Vieux-Limoilou, especially in a central position." (case 2) "Here at the Mont royal station it is pushing back." (case 3) |
| **External spatial entity_ Defined position** | SPAT2 | A spatial entity with specific location that is external to the organization concerned with the participation data. | "Lead by example as the plateau Mont royal does in Montreal (my humble opinion)" (Case2) |
| **Similar organization_ Defined position** | SPAT3 | A spatial entity with specific location that is a similar organization. Example (organization = city, the SPAT3 = another city, organization = university, SPAT3 = another university etc.) | "Strengthen the partnership with the UADB (Alioune Diop University of Bambey, Senegal)"(Case 1) "I found this box (we find it everywhere on the island of Montreal since last summer) on the edge between the sidewalk and the street. I hope one day we will see it in Tunis" (case 4) |
| **Spatial entity_ Hypothetical position** | SPAT4 | A spatial entity which does not really exist, but which may occupy a position in a spatial frame of reference in the future. Usually, it is presented as a suggestion in PP. | "I suggest setting up an outdoor skating rink that would be located between the De Koninck and Pouliot pavilions" (Case 1) "it is therefore very attractive that would have family residences on campus for student parents" (Case 1) "I suggest setting up a public square at the corner of Canardière and 4th avenue" (Case 2) |
| **Spatial entity_ approximated position** | SPAT5 | A spatial entity that exists but that the way that it is specified in the text does not enable to identify its spatial coordinates. | "These fertilizers can be used in the organic community garden" (Case 1) "Why is the service so slow to Côte-Vertu?" (Case 3) |
| **Cities_ provinces_ Countries** | SPAT6 | Cities, provinces, and countries cited in comments. Eventually if the organization concerned with PP is a city, other cities would be "Similar cities SPAT3 instead of SPAT6". | "By 2050, Africa will be the most populous continent and we will have to innovate to think about policies" (Case 1) "Lead by example as the plateau Mont royal does in Montreal (my humble opinion)" (Case 2) |

**Table A5.** Spatial patterns in SP data online only.

| Online Only Spatial Pattern | Code Name | Code Meaning | Illustrative Quotes |
|---|---|---|---|
| Hashtag | SPAT7 | A word or phrase preceded by a hash sign (#), used on social media websites and applications, especially Twitter, identifying a spatial entity. This spatial entity could be of type (SPAT1, SPAT2, SPAT3, SPAT5, SPAT6) | "#Marsa Can you transform your villa into a 3-storey building with ten apartments? Anarchic Construction !! La MARSA Here is a building under construction in the city of Ezzahira La Marsa: 14 rue de l'Océan pacifique Marsa Erriadh" (case 4) |
| Location Stamp | SPAT8 | A fixed place that is restricted through spatial coordinates and represented through a GIS (e.g., a specific location with the red stamp on google map) | "Hello @amt_info. Would there be paving of the Vaudreuil station parking lot in the near future? A real field of mud. Gare de Vaudreuil" (case 3) |

**Table A6.** Temporal patterns in both online and offline SP data.

| Temporal Constructs | Code Name | Code Meaning | Illustrative Quotes |
|---|---|---|---|
| **Past absolute calendar expression (CE)** | TEMP1 | CE indicating an absolute "date" or "duration" in the past | "this is interesting because the 2016 Nobel Prize in Physics readily admits" (Case 1) <br> "That of 2014, the Limoilou in the street, having cost $20,000, I fear the amount that will be invested." (Case 2) |
| **Future absolute CE** | TEMP2 | indicating an absolute "date" or "duration" in the future | « Could this offer a trial for a year starting in September 2018" (Case 1) <br> "… to the impact of big data artificial intelligence of all these elements that will ensure that by 2022" (Case 1) |
| **Deictic CE** | TEMP3 | CE requiring knowledge of the date the commentary was drafted. The date the commentary was written should be known. | "We don't even count the number of outages on the orange line since the start of the year @stminfo, compensation for subscribers?" (Case 3) <br> "two weeks ago through the governorate order office for immediate cancellation of the closure order" (Case 4) |
| **Textual anaphoric CE** | TEMP4 | CE whose temporal antecedent must be found in the commentary. | "there is an initiative that was launched in 2013 3 years after the start of this initiative" (Case 1) <br> "today, I went to the toll at 7:44 and I paid in cash the lady hands me a ticket from a subscriber who went 10 min before me" (Case 4) |
| **Founded anaphoric CE** | TEMP5 | CEs based on the knowledge of the world. All these expressions can correspond to a specific "date" or "duration" in the past. | "we shouldn't do a bit like in the 1950s with programs aimed at … " (Case 1) <br> "after having explored, buildings dating from the 15th to the 19th century" (case 4) |

**Table A7.** Temporal Patterns in SP data online only.

| Online Only Spatial Pattern | Code Name | Code Meaning | Illustrative Quotes |
|---|---|---|---|
| Temporal Hashtag | TEMP6 | A word or phrase preceded by a hash sign (#), used on social media websites and applications, especially Twitter, identifying a temporal entity. This temporal entity could be of type TEMP1 or TEMP2. | "You women, you charm it … Happy Women's Day and THANKS 🌷 Far from the public debate … Today is #8March, an exceptional day" (case 4) |

## Appendix C

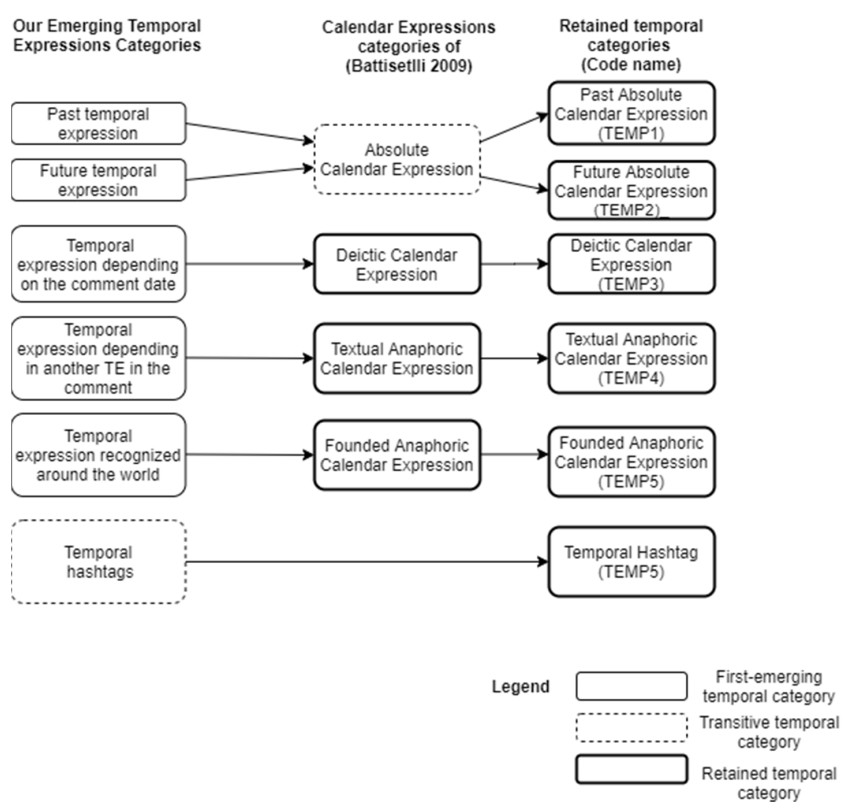

**Figure A1.** Adaptation of temporal patterns to the categorization of (Battistelli 2009) (Calendar expressions categories of (Battistelli 2009) coulb be found in [28]).

## Appendix D

**Table A8.** Correlations detected between patterns.

| Correlated Patterns | SPPs Comment | Patterns |
|---|---|---|
| **Suggestion** (SEM2) And **SE with hypothetical position** (SPAT4) | *"I suggest setting up an outdoor skating rink that would be located between the De Koninck and Pouliot pavilions"* (Case 1) | SEM2: « *I suggest* » SPAT4: « *skating rink that would be located between the De Koninck and Pouliot pavilions"* |
| | *"I suggest setting up a public square at the corner of Canardière and 4th avenue"* (Case 2) | SEM2: « *I suggest* » SPAT4: « *public square at the corner of Canardière and 4th avenue"* |
| | « I think that the university must present itself as a society in itself . . . I think it would be very attractive to have family residences on campus for student parents." (Case 1) | SEM2: « *I think it would be* » SPAT4: « *family residences on campus* » |
| | *"I suggest to build a biogas 'mini-factory' on campus in order to valorize all residual materials and produce sustainable fertilizers at the same time"* (case 1) | SEM2: « *I suggest to* » SPAT4: « *a biogas mini factory* » on compus » |

**Table A8.** *Cont.*

| Correlated Patterns | SPPs Comment | Patterns |
|---|---|---|
| **Suggestion** (SEM2) And **Future absolute calendar expression** (TEMP2) | « *Could this offer a trial for a year starting in September 2018?*" (Case 1) | SEM2: « *Could this* » TEMP2: « *in september 2018* » |
| | "*I suggest not waiting until 1 March 2017 and starting tomorrow morning to take the reusable baskets when going to the supermarket to do the shopping. This civic approach will prevent us from throwing away 300,000,000 plastic bags annually.*" (case 3) | SEM2: « *I suggest* » TEMP2: « *1 March 2017* » |
| | "*it would be interesting to think about the impact of the artificial intelligence of big data of all these elements which will ensure that by 2022 it is clear that if we do nothing, we will have outdated graduates*" (case 1) | SEM2: « *It would be interesting* » TEMP2: « *by 2022* » |

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
