# Peer review of "The Identification of Stakeholders’ Living Contexts in Stakeholder Participation Data: A Semantic, Spatial and Temporal Analysis"

_land, doi:10.3390/land11060798_

Round 1
Reviewer 1 Report
The paper is interesting. In it the authors offer an interesting point of view on the issue of the "stakeholders' living context identification” highlighting how stakeholders implicitly identify their living contexts in their SP comments.
They proposed a qualitative analysis of SP data from four case studies in two countries.
The theme of the research is consistent with that of the Special Issue entitled “Place-Based Urban Planning”.
They identify three set of semantic, spatial and temporal patterns aimed at capture the stakeholders’ living contexts in SPPs data.
Overall, the paper is fairly well structured. There are several organisational problems with regard to the following issues:
- In Subsection 3.2. it is necessary to introduce the case studies and areas of analysis in order to understand Appendix A.
- In subsection 3.3.1, since it is part of the Methodology section, it is necessary to introduce the “structured multi-steps approach”.
- In subsection 3.3.1 it is necessary to clarify the meaning of the period at lines 309-312 “Only temporal patterns were adapted following the model of [33] that proposed a model of four calendar expressions, which are similar to (5 out of 6) the temporal patterns we identified in data (see appendix D)”, in particular as regards (5 out of 6).
- In this sub-paragraph, the authors recall Appendix D before the other Appendices, which creates confusion, since the order of appearance of the Appendices should be the order in which they are presented in the text of the paper.
- The quality of Figure 2 should be improved, in it the text is very blurry.
- In section 4, Appendix B has been invoked, namely after D.
- The Discussion and Conclusion section should be two separate, as the first section is very important to highlight some summary comments on the results obtained.
- The statement quoted at lines 785-788 “First, future research could investigate the possibility to implement artificial intelligence algorithms to automatically apply the semantic, spatial and temporal contextualization approach through automatic identification of patterns in SP data”, is not entirely shareable, as in the literature there are several algorithms to support analysis of this type, as well as several IT support. The authors said that I do not want to propose a reconnaissance of the IT tools to support SPP, but as for the algorithms, they should call some for consistency.
Author Response
Dear Reviewer 1,
Please see the attachment.
Best regards,

Reviewer 2 Report
I feel that the issue of the paper is exciting. But there are some points to be revised.
- There are too many subsections in the literature review section. How about combining some of them?
- The Data structure (figure 2) is hard to read. How about simplifying or changing into a table? And the diagram about the data analysis process will be helpful to understand your research process.
- How about merging '4. research finding' and '5. research outcome' into '4. result' as the traditional format of LAND? I cannot understand the difference between chapters 4 & 5.
- Please make the discussion section separate. And show the paper's originality by comparison with existing documents in the discussion section.
Author Response
Dear Reviewer 2,
Please see the attachment.
Best regards,

Round 2
Reviewer 1 Report
The authors in this new version of the paper have solved the problems found in the first version.
The quality of the paper has now improved and so it can be considered accepted for publication in Land.
Reviewer 2 Report
I feel that the paper has been revised well. But formatting is needed. The size of the table and the figure is a little different.